# Is the Volume-of-Fluid Method Coupled with a Sub-Grid Bubble Equation Efficient for Simulating Local and Continuum Aeration?

**Lourenço Sassetti Mendes** [1,2,]*[ID]**, Javier L. Lara** [1][ID] **and Maria Teresa Viseu** [2]

¹ IHCantabria—Instituto de Hidráulica Ambiental, Calle Isabel Torres 15, 39011 Santander, Spain; jav.lopez@unican.es
² Laboratório Nacional de Engenharia Civil, Avenida do Brasil 101, 1700-066 Lisbon, Portugal; tviseu@lnec.pt
* Correspondence: lmendes@lnec.pt

**Abstract:** Air entrainment is common in free surface flows in large hydraulic structures (e.g., spillways, chutes, energy dissipation structures) and must be considered to assure an effective and safe operation. Due to the large size of the prototype structures, it is infeasible to model individual air bubbles. Therefore, using the OpenFOAM toolbox, an efficient simulation model for aerated flows is developed for engineering purposes. The Reynolds-averaged Navier–Stokes equations and the volume-of-fluid method are coupled with a sub-grid bubble population model that simulates entrainment and transport. A comprehensive assessment of the effectiveness, computational cost, and reliability is performed. Local and continuum bubble entrainment are evaluated in two distinct flows: an impinging jet and along a spillway chute. Aeration is induced, respectively, by a shear flow and by the thickening of the turbulent boundary layer. Moreover, a detailed sensitivity analysis of the model's parameters is conducted. Calibration and validation are performed against experimental and prototype data. Among the analyzed entrainment formulations, the one depending exclusively on the turbulent kinetic energy is the only applicable to different flow types. Good accuracy is found, meeting engineering standards, and the additional computation cost is marginal. Results depend primarily on the volume-of-fluid method ability to reproduce the interface. Calibration is straightforward in self-aeration but more difficult for local aeration.

**Keywords:** local aeration; free-surface aeration; volume-of-fluid; sub-grid bubble equation; hydraulic structure

## 1. Introduction

The numerical modelling of aerated flows in hydraulic structures is exceptionally complex. The physical processes are not entirely understood and the most accurate models require exorbitant computational resources. Therefore, a reliable and efficient simulation tool for engineering purposes is of utmost importance. A solver based on the Reynolds-averaged Navier–Stokes equations (RANS) and the volume-of-fluid method (VOF) is coupled with a sub-grid bubble population model (SGBM). The bubbles entrainment and transport are only simulated by the SGBM. Hence, the RANS model does not require an extremely high spatial resolution to capture the bubbles. The sub-grid models are conceived to simulate physical processes where the spatial resolution is not sufficient [1]. This framework is characterized by the stability and efficient free-surface tracking of the RANS and VOF and the speed enhancement provided by the SGBM. For practical applications, the air entrainment formulation must apply to different types of flow and the setup of the bubble model parameters should be straightforward. Hence, proper evaluation of this framework's advantages and liabilities is required.

Water flows in hydraulic structures are frequently characterized by high velocity, high turbulence and large quantities of air exchanged at the free-surface. Regions of two-phase

air–water flows are observed in many applications, including spillways, slope channels, weirs, impinging jets, aeration mixing tanks, energy dissipation structures and wave breaking [2–4]. Kobus [5] declared that: "for many hydraulic structures, safe operation can only be achieved if not only the characteristics of the water flow are considered, but due attention is also given to the simultaneous movement of air in the system".

Effects of aeration in hydraulic engineering include flow bulking, drag reduction, pressure peak modification, interaction with the turbulence field, re-oxygenation and transfer of atmospheric gases [3]. Bulking demands larger sections, and inside conduits may cause the transition to the pressurized flow regime. The small air bubbles behave as rigid spheres at the flow boundaries, reducing drag and increasing flow momentum. Furthermore, air concentrations ranging from 1 to 8% reduce or suppress cavitation and associated problems that may occur in high-velocity flows [6–9].

When in large number, bubbles enhance atmospheric gas exchanges by a substantial enlargement of the interface area [8,10]. At large dams, self-aeration on spillways is crucial for downstream ecology [7]. Both low and high dissolved air concentrations are harmful to the ecosystem. Industry uses bubble plumes and jets for mixing and aeration due to buoyancy-driven currents [2]. Bubble presence alters the turbulence levels, and thus the energy dissipation rates. Breakup and coalescence, which depend on turbulence, also affect the turbulent kinetic energy production [2,11,12].

Air entrainment is the incorporation of air bubbles in water [13]. Spatially, it can be localized or continuous. Local aeration occurs at plunging jets, waves and hydraulic jumps, particularly at the discontinuity between the impinging jet and the surrounding waters. In continuous aeration, the air entrainment occurs at an extensive region of the air–water interface, generally parallel to the flow direction, as observed in spillways [8]. Until today, some of the physical mechanisms involved are not entirely understood, especially regarding the turbulent interactions [14]. After onset, air packets with a wide range of sizes (e.g., pockets, droplets, bubbles) are transported and suffer complicated processes such as fragmentation, coalescence, diffusion, dissolution, pressure-induced volume change, and buoyant degassing [2,11,15].

Aerated flows are highly complex. Hence, they have been mainly analyzed on physical models, regardless of all the scale effects and limitations. Chanson [3] remarks that "the modelling of aerated flows is presently restricted by the complexity of theoretical equations, some limitations of numerical techniques, a lack of full-scale prototype data and very-limited detailed experimental data sets suitable for sound CFD [(Computational fluid dynamics)] model validation". An integrated physical and numerical modelling may mitigate the time, costs, and applicability shortcomings. The first provides unique data to validate the CFD models. While the second allows testing of several solutions in a short time, providing important insights before the construction of the laboratory setup and may overcome some scale effects presented in the physical models.

Use of computational fluid dynamics (CFD) in hydraulic structures has increased over the past two decades, supported by technological advances. However, it is not established in practical engineering. CFD allows the rapid testing of innovative designs with significant cost saving and provides a detailed analysis of the results. Despite many challenges, CFD modelling has enormous potential for research and practical applications.

Numerical modelling of air–water flows is exceptionally difficult. It can be done by a single-phase with averaged properties if the air concentration is small (<15%). Greater air concentrations demand that air and water are treated as separate phases [2,3]. Ideally, air entrainment, bubble formation, transport, transformations, dissolution and atmosphere restitution should be considered. Furthermore, various critical difficulties arise: the phases interface is usually hard to determine [16,17]; the range of length scales is huge in extent, i.e., Kolmogorov length, bubble diameter, surface roughness, turbulence eddies and large characteristic lengths; the coupling of the equations of the velocity, pressure, volume fraction and turbulence of the phases is extremely difficult [2,3].

Several methods were developed to model air–water flows. Direct Numerical Simulations (DNS) demand exorbitant computational costs, and thus are restricted to applications focused on particular bubble processes [11,17]. In the Lagrange–Lagrange method, both air and water are represented by Lagrangian particles. In the Euler–Lagrange method, water is solved in a Euler referential. Bubbles are represented by a discrete phase of Lagrangian particles, which is limited to a maximum concentration of approximately 15%. Both methods require millions of particles that imply enormous computational effort [17,18]. The complete two-phase Euler models are appropriate to simulate most aerated flows, especially with high air concentrations (>10–20%). However, they demand very high computational efforts and have complex frameworks for fluids interactions [2,18]. The mixture model solves a continuous single-phase of volume-averaged properties of both fluids in a Eulerian grid and has no interface tracking; therefore, it is only applied when the interaction of phases is not clear [18]. Interface models, such as the volume-of-fluid (VOF) or level-set (LS), are meant for two or more immiscible fluids with a precise interface and distinct densities. Although each fluid has a continuity equation, the method is based on a single-phase flow in a Eulerian grid. Only one set of the Navier–Stokes equations is solved for a mixture phase with volume-averaged properties. Additionally, the free-surface is determined by an efficient tracking technique, which tends to separate the fluids. The interface models are widely applied in free-surface flows of hydraulic structures and demand low to average computational costs. The surface tension and interface interactions are difficult to calculate, and unrealistic cavity may occur [18,19]. In all Euler models, the reproduction of bubble geometry requires exceptionally high mesh resolutions; hence, simulations are limited to very reduced domains.

One approach to simulate aerated free-surface flows is to combine an interface model for air and water with a specific model for the bubble dynamics. This provides the high efficiency of free-surface tracking and also restricts the range of time and length scales to be modelled. Therefore, the resolution of the interface model is optimized for the main flow, which reduces the domain's number of elements and the computational cost [11,18].

For instance, an interface model can be combined with a mixture or Euler model for the bubble flow. However, both phases' momentum equations are challenging to match, and the determination of free surface boundary conditions is complex [17,20,21]. A second approach proliferates in hydraulic structures: coupling an interface model with a sub-grid bubble density model that depends on local flow properties such as turbulence and velocity. This approach relies on accurate entrainment formulations, which may not suit the diverse type of flows. Based on a single-phase model, they are limited to bubble volume concentrations not exceeding 10 to 20%. Moreover, a mass transfer between the air and bubble phases to account for free-surface exchanges should be considered, increasing the complexity [2,18,22].

Significant developments to the combination of interface models with sub-grid bubble equations emerged in the last decade. High expectations were created for the future. Most applications use Reynolds Averaged Navier–Stokes equations (RANS), a uniform-size bubble and an entrainment formulation based on turbulence. They depend on parameter calibration and do not consider bubble diffusion due to turbulence. The following works are considered the most relevant advances in hydraulic engineering. Shi et al. [11] simulated the evolution of a multiple-size bubble population in a breaking wave event. Entrainment is based on the strain rate, and bubble breakup and coalescence are included. The turbulence model accounts for bubble-induced effects and is responsible for bubble diffusion. Ma et al. [17] developed a method to predict air entrainment—based on the turbulent kinetic energy—and the transport of uniform size bubble in a plunging jet and a hydraulic jump. Ma et al. [23] studied a hydraulic jump with both RANS and DES (Detached Eddy Simulation) turbulence models. Valero et al. [12] analyzed the multiphase flow in a USBR type II stilling basin. Xiang et al. [24] framework comprises a multiple-size bubble and is applied to hydraulic jumps. Lopes et al. [16] used an explicit term based on Ma et al. [17] to determine air entrainment in a plunging jet. Lopes et al. [25] studied

self-aeration in a stepped channel. The latter proposed an entrainment formulation that does not require calibration. Moreover, a mass transfer between the air and bubble phases is tested.

Typically, the large dimensions of hydraulic structures and the flow velocity make a mesh that reproduces the individual air bubble and pockets unfeasible. An excessive number of cells would be required to apply the volume-of-fluid method (VOF) properly. The combination of a VOF with a sub-grid bubble model overcomes this limitation. This approach requires a mesh fine enough to reproduce the free surface, yet enough coarse to solve the entrained bubbles exclusively with the sub-grid model [19].

The state of the art research on sub-grid bubble models coupled to VOF-based solvers generally focuses on a single flow type (e.g., plunge jet, open channel, closed conduit, hydraulic jump). Therefore, it is necessary to identify a method that meets local and continuous aeration. Most aforementioned works are based on numerical reproduction of laboratory experiments at Froude's law scale, while entrainment and bubble interactions are strongly related to Weber and Reynolds numbers [26], which involve important scale effects. In addition to all the technical difficulties in measuring entrained air, similarity can only be achieved in large installations. Prototypes are the ultimate source of data; however, collection requires devices that can withstand extreme velocities, and campaigns are very expensive and difficult to covenant with infrastructure owners [3]. Moreover, there is no sensitivity analysis to model parameters, possibly involving a very complex calibration process prior to each application.

The main goal of the present work is to develop an accurate and robust numerical model for practical applications of aerated flows, meeting a range of hydraulic engineering purposes. The model must comply with local and continuum aeration and have a computational cost similar to the interface models. Moreover, the calibration effort for each application must be small.

Local and continuum bubble entrainment are studied with an impinging jet and a spillway chute. Hence, two prevalent, though distinct, onset mechanisms, which are the basis for more complex air-entrained flows are covered. In the first case, aeration is due to the shear flow generated by a vertical jet penetrating a pool at rest. In the second, the rise of the bottom turbulent boundary layer destabilizes the free-surface and promotes the entrainment of air pockets. Calibration and validation are performed against laboratory data of Chanson and Manasseh [15] and prototype-based analytical data of Chanson [27].

A sub-grid bubble population model (SGBM) is coupled with a RANS model, which includes the VOF method, to evaluate aerated water fluxes in hydraulic structures. Based on Shi et al. [11], the sub-grid model simulates the entrainment and transport of multiple-sized bubbles divided into clusters. Bubbles' diffusion is regulated by turbulence that considers bubble-induced effects. This framework is distinguished by the efficient simulation of a bubble population, comprising significant bubble processes, in a reliable interface model.

Three different formulations of bubble entrainment based on turbulence and strain rate in two reference flows are evaluated. The mesh dependence and efficiency of the coupled model are evaluated. In addition, a rare but fundamental sensitivity analysis of various model parameters demonstrates the accuracy and robustness of practical applications.

The SGBM equations are proposed by Shi et al. [11]. The two additional entrainment formulations follow the same equation of the previous author, but use the production fields presented by Ma et al. [17] and Lopes et al. [25] (based on Hirt [28]). The methodology and the numerical implementation are original and developed exclusively for this research. Several additional methods were necessary (Appendix A). An extensive library has been created in the open source OpenFOAM® software toolbox [29], which has a structured code and a large user community, which favours the inclusion of new resources.

The paper is organized as follows. First, the mathematical models are presented. Next, validation and calibration are performed in the impinging jet and the spillway chute. Finally, the discussion and conclusions are presented.

## 2. Mathematical Models

### 2.1. Base Model

A sub-grid bubble population model (SGBM) is combined with *interFoam*, a Reynolds-Averaged Navier–Stokes (RANS) equations solver for two incompressible, isothermal and immiscible fluids. This solver is included in the OpenFOAM® toolbox version 1906 [29]. The interface capturing is based on a volume-of-fluid (VOF) approach that incorporates an interfacial compression flux term [25,30–32].

Mass (1) and momentum conservation (2) equations are solved for a single-phase that includes two fluids, therefore sharing the same velocity field, water and air.

$$\nabla \cdot \boldsymbol{V} = 0 \tag{1}$$

$$\frac{\partial \rho \boldsymbol{V}}{\partial t} + \nabla \cdot (\rho \boldsymbol{V}\boldsymbol{V}) = -\nabla p^* - \boldsymbol{g}\boldsymbol{X} \cdot \nabla \rho + \nabla \cdot (2(\mu + \mu_t)\boldsymbol{S}) + \boldsymbol{f}_\sigma + \boldsymbol{f}_b \tag{2}$$

where $\boldsymbol{V}$ is the RANS velocity vector, $\rho$ is the density, $t$ is the time, $p^*$ is the pseudo-dynamic pressure, $\boldsymbol{g}$ is the gravitational acceleration, $\boldsymbol{X}$ is the position vector, $\mu$ is the molecular dynamic viscosity, $\mu_t$ is the eddy viscosity coefficient (i.e., turbulent dynamic viscosity) and $\boldsymbol{S}$ is the strain rate tensor. The surface tension force term is defined by Equation (3).

$$\boldsymbol{f}_\sigma = \sigma \kappa \nabla \alpha \tag{3}$$

where $\sigma$ is the surface tension, $\kappa$ is the curvature of the interface and $\alpha$ is the water volume fraction.

The partial coupling between the VOF and SGBM is accomplished by a bubble buoyancy force term (4). This force is only applied to the water fraction, assuring that an air pocket immersed in water is not subject to bubble buoyancy. Otherwise, these concurrent sources would overdo the buoyancy.

$$\boldsymbol{f}_b = -\rho \alpha C_b \boldsymbol{g} \tag{4}$$

where $C_b$ is the bubble volumetric concentration.

Particularly using the VOF technique, a single-phase scalar function ($\alpha$) is used to track the interface. $\alpha$ defines the fluids volume fraction in each cell and ranges from 0 to 1. If $\alpha = 1$, the cell is full of water, and when $\alpha = 0$ it is full of air. Other $\alpha$ values identify interface cells. This phase advection is described by Equation (5).

$$\frac{\partial \alpha}{\partial t} + \nabla \cdot \boldsymbol{V}\alpha + \nabla \cdot \boldsymbol{v}_c \alpha(1 - \alpha) = 0 \tag{5}$$

$$\boldsymbol{v}_c = C_\alpha |\boldsymbol{V}| \frac{\nabla \alpha}{|\nabla \alpha|} \tag{6}$$

An artificial compression preserves a sharp interface between air and water (third term of Equation (5)). A compression velocity (6) proportional to the velocity field magnitude is applied perpendicular to the interface. $C_\alpha$ is a coefficient that triggers the interface compression term and usually ranges from 0 to 1.

Any VOF phase property $\Phi$ (e.g., $\rho$, $\mu$) is a volume-average of the intrinsic fluid property of water and air (7).

$$\Phi = \alpha \Phi_{water} + (1 - \alpha)\Phi_{air} \tag{7}$$

### 2.2. Sub-Grid Bubble Model

The bubble modelling is based on Shi et al. [11] who simulated air-bubble plumes' evolution in a breaking wave. The sub-grid bubble model (SGBM) comprises entrainment, advection and turbulence-induced diffusion of a multiple-size bubble population. Coalescence and break-up methods are included, although not used in the presented work.

The bubble population is split into groups based on radius. $\eta_g$ is the number of groups adopted. Each group, designated by index $i$, is governed by one transport equation, which includes inter-group adjustments due to break-up and coalescence. The transport equations are written in terms of bubble number concentration, i.e., the number of bubbles of each group per volume unit, facilitating the inter-group adjustments. A bubble buoyancy-force term is added to the RANS momentum Equation (2), and bubble-induced effects are considered in the turbulence model. Nonetheless, both dissolution of bubbles in water and inter-group adjustments caused by bubble size changes due to pressure variation are neglected. This first-stage development cannot reproduce bulking, and the volumetric bubble concentration must not exceed 10 to 20% [2].

The implementation in OpenFOAM® requires several additional methods (see Appendix A).

### 2.2.1. Bubble Entrainment

A method is developed to locate the bubble entrainment. It occurs at a single layer of cells just below an iso-surface defined by a specified VOF fraction value ($\alpha_{ent}$). The steps to determine the entrainment layer are presented in Appendix B.

The potential bubble entrainment (8) is proportional to a production term based on flow properties ($P_r$) and is only active if a threshold ($P_{r0}$) is exceeded.

$$E_{n,i}^P = a_b \, P_r \, F_{b,i}, \quad P_r > P_{r0} \tag{8}$$

where $a_b$ is a coefficient to calibrate and $F_{b,i}$ is the bubble number concentration distribution factor of group $i$.

It is of utmost importance to find a production term accurate for both local and continuum aeration. Hence, three approaches of the most prominent work on sub-grid bubble models combined with an interface model (VOF, Level-Set) are evaluated. They are based on turbulence yet on different properties and were proposed for distinct flows.

Shi et al. [11] consider that the bubble entrainment is directly proportional to the turbulent kinematic viscosity ($\nu_t$) and the square of the magnitude of the strain rate tensor ($|S|^2$), which is a function of the velocity gradients, as follows:

$$P_r^S = \nu_t |S|^2 \tag{9}$$

This production term (9), defined by superscript 'S', is related to the production of turbulence kinetic energy (24) and was applied to a breaking wave event.

Ma et al. [17] simulated aeration in a plunging liquid jet and a hydraulic jump and proposed a production term directly proportional to the turbulent kinetic energy ($k$), defined by superscript 'K', as follows:

$$P_r^K = k/g \tag{10}$$

Hirt [28] proposed a bubble onset production term which is a function of $k$ and the turbulent dissipation rate ($\varepsilon$). This approach applies the theoretical concept of turbulent length scales to estimate the surface disturbances as a function of the turbulent eddies. More recently, Lopes et al. [25] tested a similar approach to model aeration in a stepped spillway. This term, defined by superscript 'KE', can also be proportional to $\nu_t$ and $k$, as follows:

$$P_r^{KE} = k^{3/2}/\varepsilon \; \propto \; \nu_t \, k^{-1/2} \tag{11}$$

In the entrainment, the bubble population size distribution must be defined. According to Deane and Stokes [33], this distribution is related to the Hinze scale, which is associated to the turbulent dissipation rate and the surface tension. Chanson [8] suggests that bubble size distributions in turbulent shear flows are best fitted by a log-normal distribution, although both Gamma and Weibull distributions are also satisfactory. Shi et al. [11] recommend that the bubble groups' average radius be equally spaced in the logarithmic

scale. Xiang et al. [24] found that ten equally sized spaced bubble groups are sufficient to resolve the bubble size evolution over the entire range (0 to 10 mm).

The bubble number concentration distribution factor of each bubble group ($F_{b,i}$) represents the number of bubbles entrained from that group in respect to other groups. $F_{b,i}$ is determined according to the chosen bubble size distribution function. Furthermore, to ensure that the volumetric onset is independent of the selected population characteristics (number of groups and size), the $F_{b,i}$ factor is normalized for a population with a total arbitrary reference volume of one cubic meter, as follows:

$$\sum F_{b,i} vol_{b,i} = 1 \ m^3 \tag{12}$$

where $vol_{b,i}$ is the volume of a single bubble with the average radius of group $i$.

The local bubble number concentration must not exceed a maximum bubble fraction (e.g., $C_{b,max} \leq 1$). Hence, the available capacity for entrainment of each group is determined by:

$$N_{b,i}^{capacity} = max\{C_{b,max} \ F_{b,i} - N_{b,i}, 0\} \tag{13}$$

where $N_b$ is the bubble number concentration.

The effective entrainment equals the potential entrainment source (8), despite being limited to assure that the local $C_{b,max}$ is not surpassed, as follows:

$$E_{n,i} = min\{E_{n,i}^P, \ N_{b,i}^{capacity}/\partial t\} \tag{14}$$

### 2.2.2. Bubble Transport

A bubble transport equation is defined using the bubble number concentration ($N_{b,i}$) (15) for each bubble group $i$ as follows:

$$\frac{\partial N_{b,i}}{\partial t} + \nabla \cdot (N_{b,i} \boldsymbol{V}_{b,i}) = \nabla \cdot (D_b \nabla N_{b,i}) + E_{n,i} + S_{n,i} \tag{15}$$

where $S_n$ is the source/sink term from coalescence and breakup. This equation is defined for $\eta_g$ groups.

The bubble advection velocity of a group (16) is determined by adding a vertical slip velocity (17) to the base model velocity field. The slip velocity formulation of Clift et al. [34]'s is applied considering the average group radius, as follows:

$$\boldsymbol{V}_{b,i} = \boldsymbol{V} + w_{s,i} \ \boldsymbol{K} \tag{16}$$

$$w_{s,i} = \begin{cases} 4474 \ r_{b,i}^{1.357}, & 0 \leq r_{b,i} \leq 7 \times 10^{-4} \\ 0.23, & 7 \times 10^{-4} \leq r_{b,i} \leq 5.1 \times 10^{-3} \\ 4.202 \ r_{b,i}^{0.547}, & r_{b,i} > 5.1 \times 10^{-3} \end{cases} \tag{17}$$

where $\boldsymbol{K}$ is the unit vertical vector and $r_b$ is the bubble radius.

Bubble diffusivity (18) can be expressed as proportional to $\nu_t$ and inversely proportional to the Schmidt number for gases in water ($S_g$).

$$D_b = \frac{\nu_t}{S_g} \tag{18}$$

After solving the transport equations of the groups, the total volumetric bubble fraction (19) is calculated by summing the volume of all bubbles.

$$C_b = \sum_i N_{b,i} \ vol_{b,i} \tag{19}$$

The transport equation of each bubble group (15) is solved independently. Hence, there is a possibility that the resulting $C_b$ exceeds the maximum threshold ($C_{b,max}$). If a cell

surpasses this limit, a bubble number artificial correction is applied to all groups: $N_{b,i}$ is bounded (20) and $C_b$ is recalculated (19).

$$N_{b,i}^{bounded} = \frac{N_{b,i}}{max\left\{1, \frac{\Sigma_i C_{b,i}}{C_{b,max}}\right\}}$$ (20)

Afterwards, bubbles are detrained (i.e., eliminated) where $\alpha$ is lower than a specified threshold ($\alpha_{det}$), which must be lower than $\alpha_{ent}$.

### 2.3. Turbulence Models

Since two types of air entrainment processes are simulated, two different turbulent models are used. The $k$–$\varepsilon$ turbulence model is conceived for internal flows and is the most common in RANS simulations; therefore, it is applied to the impinging jet. The $k$–$\omega$ SST is used in the spillway due to its advantage for boundary flows. Both models have two transport equations that are used to calculate the eddy viscosity: one for the turbulent kinetic energy ($k$) and another for the turbulent dissipation rate ($\varepsilon$) or the turbulent specific dissipation rate ($\omega$). In *interFoam* these models do not include density explicitly; hence the kinematic eddy viscosity ($\nu_t$) is computed (23) instead of the dynamic form ($\mu_t$) [35]. The standard coefficients are utilized.

The standard model of Launder and Spalding [36] is the base of the $k$–$\varepsilon$ model, which is defined by Equations (21) and (22).

$$\frac{\partial k}{\partial t} + \nabla \cdot (\boldsymbol{V}k) = \nabla \cdot \left[\left(\nu + \frac{\nu_t}{\sigma_k}\right)\nabla k\right] + G - \varepsilon + S_{k,b}$$ (21)

$$\frac{\partial \varepsilon}{\partial t} + \nabla \cdot (\boldsymbol{V}\varepsilon) = \nabla \cdot \left[\left(\nu + \frac{\nu_t}{\sigma_\varepsilon}\right)\nabla \varepsilon\right] + C_1 G \frac{\varepsilon}{k} - C_2 \frac{\varepsilon^2}{k} + S_{\varepsilon,b}$$ (22)

$$\nu_t = C_\mu k^2 / \varepsilon$$ (23)

$$G = 2\,\nu_t |\boldsymbol{S}|^2$$ (24)

where $\nu$ is the kinematic viscosity, $G$ is the turbulent kinetic energy production due to the mean velocity gradients, $S_{k,b}$ and $S_{\varepsilon,b}$ are bubble source terms. $\sigma_k = 1.0$, $\sigma_\varepsilon = 1.3$, $C_1 = 1.44$, $C_2 = 1.92$ and $C_\mu = 0.09$ are coefficients.

The $k$–$\omega$ SST turbulence model is based on Menter et al. [37] and follows Equations (25) and (26).

$$\frac{\partial k}{\partial t} + \nabla \cdot (\boldsymbol{V}k) = \nabla \cdot [(\nu + a_k \nu_t)\nabla k] + P_k - \beta^* \omega k + S_{k,b}$$ (25)

$$\frac{\partial \omega}{\partial t} + \nabla \cdot (\boldsymbol{V}\omega) = \nabla \cdot [(\nu + a_\omega \nu_t)\nabla \omega] + \gamma P_\omega - \beta \omega^2 + 2(1 - F_1)a_{\omega 2}\frac{\nabla k \cdot \nabla \omega}{\omega} + S_{\omega,b}$$ (26)

$$\nu_t = a_1 \frac{k}{max(a_1 \omega, b_1 F_{23}\boldsymbol{S})}$$ (27)

$$P_k = min(G, c_1 \beta^* k\omega)$$ (28)

$$P_\omega = min\left[\frac{G}{\nu_t}, \frac{c_1}{a_1}\beta^* \omega\, max\left(a_1 \omega, b_1 F_2 \sqrt{2|\boldsymbol{S}|^2}\right)\right]$$ (29)

where $P_k$ and $P_\omega$ are production terms. $S_{\omega,b}$ and $S_{\omega,b}$ are bubble source terms. $\beta^* = 0.09$, $a_1 = 0.31$, $b_1 = 1.0$, $c_1 = 10.0$, $F_2$, $F_{23}$ are coefficients. $a_k$, $a_\omega$, $a_{\omega 2}$, $\beta$ and $\gamma$ are a blend of inner and outer coefficient values using the $F_1$ coefficient.

Bubble-induced turbulent effects are considered according to Kataoka and Serizawa [38] approach. They proposed two source/sink terms, (30) and (31), for the $k$–$\varepsilon$ model. The term (31) is adapted to the $k$–$\omega$ SST model (33), assuming the relation between $\omega$ and $\varepsilon$ (32).

$$S_{k,b} = C_k C_b \frac{\nabla p^* \cdot w_s}{\rho} \tag{30}$$

$$S_{\varepsilon,b} = C_\varepsilon \frac{\varepsilon}{k} S_{k,b} \tag{31}$$

$$\varepsilon = \beta^* \omega k \tag{32}$$

$$S_{\omega,b} \approx \frac{1}{\beta^* k} S_{\varepsilon,b} \approx C_\omega \frac{1}{\beta^* k} \frac{\varepsilon}{k} S_{k,b} \approx C_\omega \frac{\varepsilon}{\beta^* k^2} S_{k,b} \approx C_\omega \frac{\omega}{k} S_{k,b} \tag{33}$$

where $C_k = 1.0$, $C_\varepsilon = 1.0$ and $C_\omega = 1.0$ are coefficients.

## 3. Results

### 3.1. Local Aeration in a Impinging Jet

The first case is employed to assess the model in localised air entrainment, which typically occurs due to geometry discontinuities or impinging flows that enhance velocity gradients, turbulence and shear layer flows.

An experiment of a vertical impinging jet in a water pool at rest by Chanson and Manasseh [15] is replicated numerically to validate the RANS and sub-grid bubble models combination, following other authors [16,17].

Nevertheless, the sub-grid bubble model is initially de-activated to isolate the RANS model performance under different mesh resolutions. Afterwards, with the sub-grid bubble model (SGBM) activated, the three bubble production term formulations—(9)–(11)—are tested. Finally, a calibration and sensibility analysis of the SGBM parameters is carried out.

#### 3.1.1. Numerical Setup

The selected setup is defined by a circular nozzle with a 0.025 m diameter ($d_N$) elevated 0.1 m above the pool. Water exits the nozzle with downward velocity ($V_N$) of 3.21 m s$^{-1}$. At the impact point, the velocity ($V_1$) is 3.5 m s$^{-1}$, the diameter ($d_1$) is 0.024 m and the jet radius ($r_1$) is 0.012 m.

The impinging jet is a complete tri-dimensional flow with no symmetry axis or planes. Around the impinging point, the water surface level oscillates randomly due to the release of entrained air. Underwater, most entrained bubbles describe a helicoidal trajectory around the jet centerline [39,40]. Nevertheless, considering a time-averaged radial symmetry of the flow and the need to perform numerous simulations to calibrate the bubble model parameters, two orthogonal mesh types are used. A set of 2D axisymmetric wedge meshes allows fast runs and a 3D mesh that reproduces the experimental setup verifies the prior set viability (see Table 1).

**Table 1.** Impinging jet—meshes.

| Name | Type | Minimum Edge Length (mm) | Total Cells |
|------|------|--------------------------|-------------|
| W8 | 2D wedge | $d_N/8 \approx 3.13$ | 28,474 |
| W16 | 2D wedge | $d_N/16 \approx 1.56$ | 41,449 |
| W32 | 2D wedge | $d_N/32 \approx 0.78$ | 57,730 |
| W64 | 2D wedge | $d_N/64 \approx 0.39$ | 74,741 |
| CS32 | 3D | $d_N/32 \approx 0.78$ | 9,916,128 |

Both mesh sets have an air layer with 0.4 m height over a 1.8 m depth water pool (Figure 1). The 2D axisymmetric meshes have a 5° wedge angle and are 0.4 m wide. The 3D mesh has 0.30 m in the $y$-axis direction (towards side-walls) and 0.80 m in the $x$-axis direction (towards outflow boundaries).

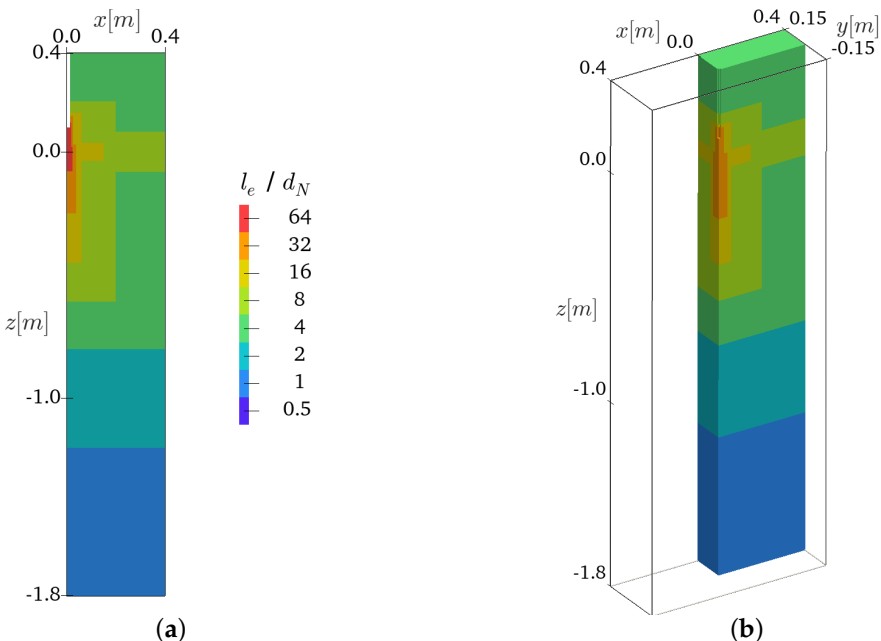

**Figure 1.** Impinging jet—mesh 2D and 3D average edge length ($l_e$): (**a**) W64. (**b**) CS32 ($x > 0$ and $y > 0$).

The same Cartesian $xyz$ coordinate system is used in all meshes. The axis origin is at the point where the jet axis intersects the water surface at rest. The $z$-axis is pointing in the opposite direction of the gravitational acceleration vector and the injector velocity.

Different cell size zones are implemented, increasing the resolution where the jet impacts the pool and air entrainment occurs. These zones are kept between 2D meshes: a higher resolution mesh is built from the previous with additional refinement. Mesh W32 mimics the 3D mesh (see Table 1). The maximum edge length is 25 mm for all meshes.

Regarding boundary conditions, both mesh sets share top, bottom and outflow definitions. The nozzle inlet has a downward velocity ($V_N$) of 3.21 m s$^{-1}$. At the top, a total pressure is combined with a binary velocity condition: zero gradient for outflow and pressure-driven normal inflow. Bottom and channel side-walls are considered walls with slip tangent velocity. At outflow boundaries, which are localized at $x = 0.4$ m (and also $x = -0.4$ m in 3D meshes), a shallow wave absorption boundary condition maintains the water level ($z = 0$ m) and absorbs the surface waves with success [41]. Consequently, the numerical domain is reduced significantly, avoiding the entire length of the laboratory channel.

The following numerical schemes are used: Crank–Nicolson time derivative; linear-upwind for the divergence of velocity, $k$ and $\varepsilon$; van Leer for the divergence of $N_b$; interface compression based on a generic limited scheme for the divergence of $\alpha$. The interface compression coefficient ($C_\alpha$) is 1.0.

Data sampling is performed after a warm-up period where semi-steady flow conditions are reached and approximate time-independence is attained for local fields and global properties, i.e., kinetic energy, turbulent kinetic energy, water level, maximum velocity, and volumetric bubble concentration. The samples are collected in each time-step and time-averaged. Using the RANS model exclusively, the sampling interval is 20 s. For the SGBM calibration, 5.0 s are sampled.

Computing the wedge mesh (W32) in an HPC cluster with an Intel Xeon E5-2680 (2.70 GHz) processor using all 16 cores, each run execution time is approximately 0.4 h per second of simulation time. A 3D mesh (CS32) run employing 16 processors (256 cores) needs 5 h per second of simulation time. When using the SGBM, calculation time increases less than 5%.

### 3.1.2. Impinging Jet Base Flow

Initially, with the sub-grid bubble model de-activated, the RANS model is assessed regarding the impinging jet's simulation under the different mesh resolutions. Jet diffusion is critical to select the optimal mesh. Thus, the following factors are helpful: the velocity profiles, the centerline velocity decay, and the bubble penetration depth.

Flow fields are analyzed at horizontal profiles ($z = \{-0.02\,\text{m}, -0.03\,\text{m}, -0.05\,\text{m}\}$) that show approximately the same behavior. Hence, the intermediate profile is presented in Figure 2. The profiles of meshes W16, W32 and W64 are proximate and have the same shape, increasing sharpness with more resolution. Apart from all the other, mesh W8 exhibits an evident incapacity to define the fields, overestimating $\nu_t$ and $k$.

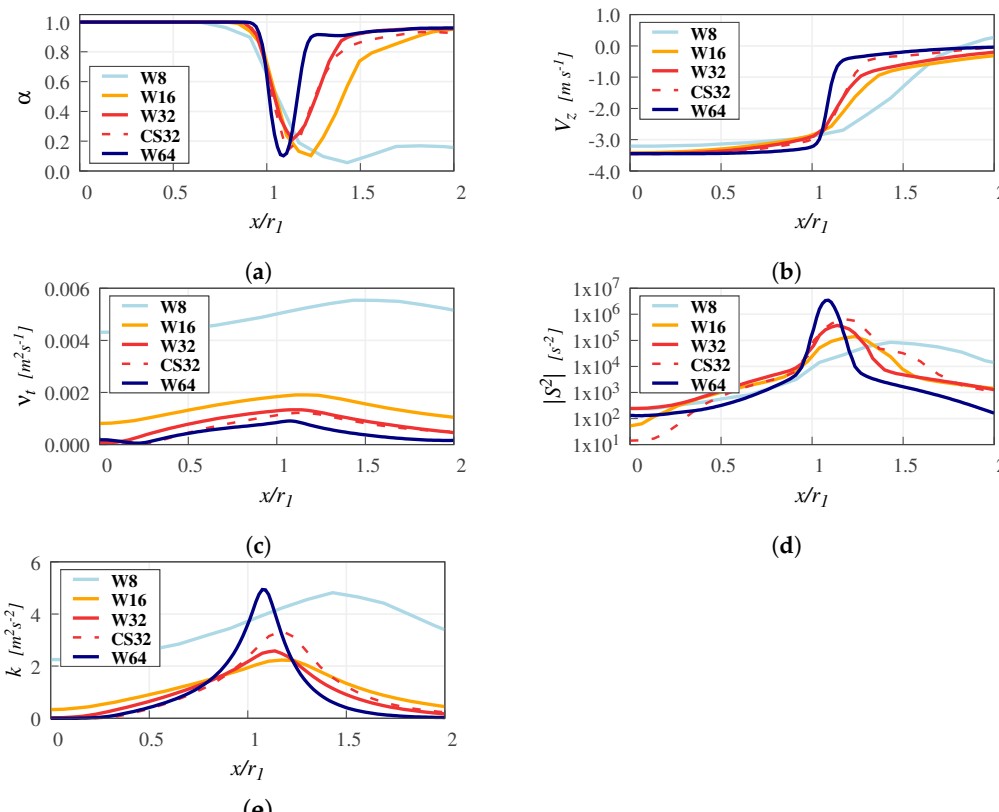

**Figure 2.** Impinging jet—mesh dependency. Profiles at $z = -0.03$ m: (**a**) $\alpha$. (**b**) $V_z$. (**c**) $\nu_t$. (**d**) $|S|^2$. (**e**) $k$.

The water volume fraction field ($\alpha$) is analyzed qualitatively at time instants because time-average can dilute the pockets and the interface shape (Figure 3). Meshes W16, W32 and W64 display a similar behavior: the jet shear layer is thin and intermittent, and the entrained air pockets ascend to the surface 5 to 15 ($r_1$) away from the jet centerline. However, in the higher resolution meshes (W32 and W64), the pockets are smaller and reach deeper. Mesh W8 shows a very wide and deep cavity in the jet shear layer zone, where most of the entrained air also exits. Furthermore, the interface thickness is exaggerated, and large surface perturbations are found in the impact point's vicinity. At the shear layer, the non-realistic entrainment of oversized air pockets is due to applying the VOF method in a mesh with an insufficient resolution to reproduce the individual bubble. According to Andersson et al. [42], in proper VOF modelling, two drawbacks arise: it requires an extraordinary mesh resolution, generally 20 cells per bubble diameter, and the surface tension forces may be overdone if the interface curvature is high. Hence, bubble and pockets tend to become spherical, and their size to be a function of the mesh resolution. Meticulous tuning of the VOF settings may suppress these pockets that perturb the flow patterns, primarily due to the buoyancy forces being overgrown.

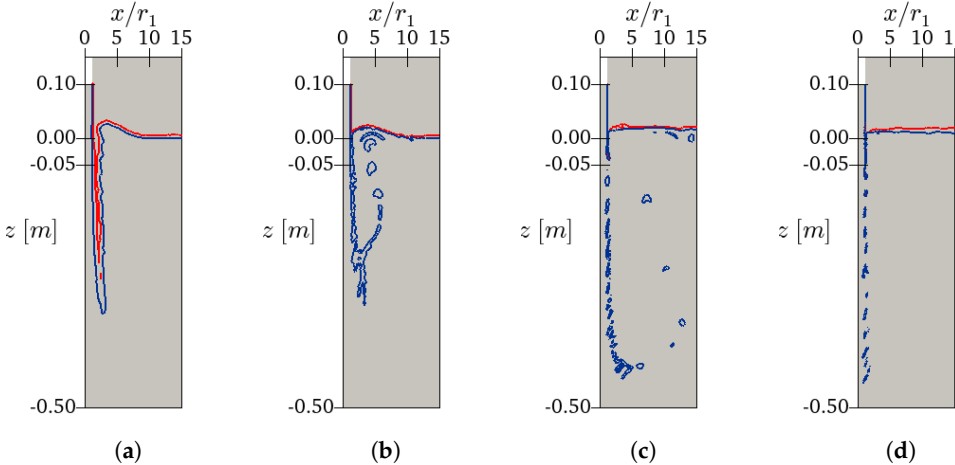

**Figure 3.** Impinging jet—2D mesh interface. VOF fraction field at $t = 60$ s. $\alpha = 0.5$ (blue), $\alpha = 0.1$ (red). Mesh: (**a**) W8. (**b**) W16. (**c**) W32. (**d**) W64.

The jet centerline velocity decay is directly related to momentum diffusion. Hence, particular characteristics of the three regions are analyzed (Figure 4). First, the developing flow region length: includes the zone between the impact point and a depth where the downward velocity of the jet core is smaller than $V_1$. Chanson [8] estimates it ranges between "5 to $10d_1$ for circular jets discharging in a fluid at rest". As shown in Figure 4, this feature is approximately matched by meshes W32 and W64. Secondly, the centerline velocity ratio in the submerged region is analyzed. The water jet in the experiment is described as "extremely smooth and transparent"; therefore, the McKeogh and Ervine [43] expression for smooth plunging jets (34) is used. Though the submerged jet region of mesh W8 follows Equation (34), a segment with a similar slope is found in other meshes.

$$V/V_1 = 3.3\big(d_1/l_j\big)^{1.1} \tag{34}$$

where $l_j$ is the jet centreline distance from the impinging point in the direction of the gravitational acceleration.

Third, Bin [44] proposed an expression to estimate the bubble penetration depth (35). Considering a bubble with $r_b = 0.5$ mm, according to (17) $w_s = 0.23$ m s$^{-1}$ and $w_s/V_1 = 0.066$.

$$h_p = 2.1 \, V_1^{0.775} \, d_1^{0.67} \tag{35}$$

With $d_1 = 0.024 \, m$ and $V_1 = 3.5 \, m \, s^{-1}$, the result is $h_p = 0.456 \, m \approx 19d_1$.

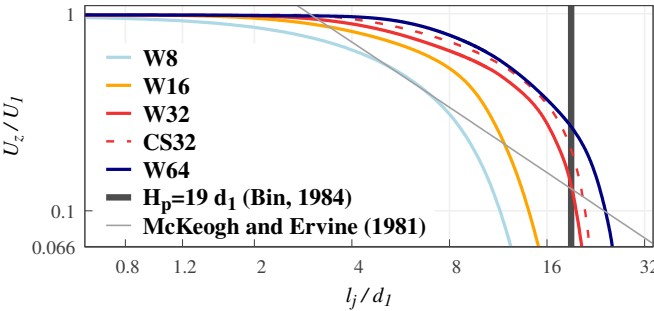

**Figure 4.** Impinging jet—mesh dependency: centerline velocity decay.

As shown in Figure 4, the coarsest mesh (W8) results are discrepant regarding the more refined meshes. With mesh W16, the air pockets in the jet shear layer zone are considered exaggerated, thus perturbing the flow patterns significantly, and the centerline velocity decay is excessive. Despite the higher resolution, mesh W64 has a good agreement

at the developing flow region length. However the penetration depth is $\approx 26d_1$, which surpasses the result of Equation (35). Even the highest resolution mesh cannot reproduce the tiny bubbles generated in the shear layer that the jet momentum diffusion induced. This fact shows one of the limitations of the VOF technique in these types of problems. Another consequence is that the exaggerated large air pockets can block part of the lateral transfer of the jet momentum. Further investigation is needed to clarify this issue and to determine the influence in bubble modelling with the VOF technique.

Overall, the 2D wedge mesh W32 (minimum cell edge length of $d_N/32$) is considered appropriate for the sub-grid bubble model calibration. Moreover, a good agreement with the 3D mesh CS32 is found at the horizontal profiles (Figure 2) and the centerline ($h_p \approx 20d_1$, Figure 4), thus validating the bi-dimensional approach. This resolution implies a very small solving time-step ($\approx 1 \times 10^{-4}$ s), demanding considerable computational efforts.

### 3.1.3. Sub-Grid Bubble Model Calibration

The sub-grid bubble model (SGBM) calibration evaluates the accuracy and sensitivity to the parameters. It is performed for three production term formulations: 'S' is based on $\nu_t$ and $|S|^2$ (9), 'K' is based on $k$ (10), and 'KE' is based on $k$ and $\varepsilon$ (11).

At this stage and as a first approximation, the focus is on the bubble entrainment proximities. Hence, inter-group transfers are neglected, and a single group bubble population ($r_b$ = 0.5 mm) is modelled.

The simulated bubble volume-fraction data ($M$) are compared with Chanson and Manasseh [15] laboratory reference data ($R$) at three horizontal profiles along $x$-axis: $z = \{-0.02$ m, $-0.03$ m, $-0.05$ m$\}$. The experimental profiles at both sides of the jet are averaged to compare against the single side 2D wedge mesh.

At each profile and to measure the capability of the model in capturing air entrainment, the integrated and the maximum bubble volume-fraction are evaluated, respectively, by a relative trapezoidal integral ($TI$) error (36) and a relative maximum error (37).

$$Integral_{err} = \frac{M_{TI} - R_{TI}}{R_{TI}} \tag{36}$$

$$Max_{err} = \frac{\max M - \max R}{\max R} \tag{37}$$

The previous errors are combined in a single value to find a good agreement in all profiles and simplify the calibration process. Hence, the accuracy criterium is the lowest value of the $IM_{err}$ error, which averages the absolute value of both error functions in the three profiles, as follows:

$$IM_{err} = \frac{1}{6} \sum_{i=1}^{3} (|Integral_{err}^{z_i}| + |Max_{err}^{z_i}|), \tag{38}$$
$$z = \{-0.02 \text{ m}, \ -0.03 \text{ m}, \ -0.05 \text{ m}\}$$

The calibration process focus on the more relevant parameters:

- $a_b$—bubble onset coefficient (8);
- $P_{r0}$—threshold for bubble onset production term (8);
- $S_g$—Schmidt number for gases in water (18);
- $\alpha_{ent}$—water volume fraction value that defines the entrainment surface (A1).

The remaining two parameters are fixed: $C_{b,max} = 1$ (maximum bubble volume fraction); $\alpha_{det} = 0.0001$ (VOF volume fraction lower-limit that triggers bubble detrainment).

More than one thousand parameter quadruplets are tested to determine the optimal combination for each production term formulation, presented in Table 2. Afterwards, the model sensitivity to each parameter is evaluated. An expeditious method avoids the trial of a tremendous number of combinations:

1. Preliminary tests point out approximate parameters ranges and roles.
2. Fixating $P_{r0}$ to a reference value, which confines the entrainment to the impinging point vicinity and $\alpha_{ent} = 0.25$, multiple combinations of $a_b$ and $S_g$ are tested. Hence, a candidate to the best quadruplet is found.
3. Finally, a variation of all four parameters $a_b$, $P_{r0}$, $\alpha_{ent}$, and $S_g$ is performed centered on the candidate to best match. If a better quadruplet is found, this step is repeated or, go back to step 2 if the deviation is excessive. Otherwise, the best quadruplet ('\*' index) is found.

The approximate number of quadruplets tested for each production term formulation is presented next. Step 1 involved the testing of more than one hundred quadruplets of multiple combinations of the four parameters. In step 2, which be could repeated, one hundred quadruplets are tested, originating from combining the ten values of both $a_b$ and $S_g$. In step 3, fifty quadruplets are tested and the step is performed one to three times.

Figure 5 shows the numerical model predictions for the bubble volumetric concentration ($C_b$) at different water depths, which are compared against Chanson and Manasseh [15] measurements. Entrainment formulations 'S' and 'K' have a very similar performance (Figure 5) and good accuracy with a $IM_{err}$ near 15% (Table 2). These two formulations perfectly reproduce profile $z = -0.03$ m. $C_{b,max}$ at profiles $z = -0.02$ m and $z = -0.05$ m is, respectively, under and over-estimated. At the deepest profile, the maximum is slightly inward; therefore, jet diffusion is lacking. Particularly in the $\alpha$, $\nu_t$, $|S|^2$ and $k$ horizontal profiles, the peak distance to the jet centerline is inversely proportional to the mesh resolution (Figure 2). This behavior exposes the VOF and turbulent models limitations to reproduce the shear region. Furthermore, this numerical setup is unable to produce acceptable results with entrainment formulation 'KE'. No parameter combination confines the bubble onset to the jet vicinity, and the air entrainment is faultily triggered at the remaining water surface.

**Table 2.** Impinging jet—calibration of sub-grid models formulations: best parameter quadruplet.

| Formulation | $a_b{}^*$ | $P_{r0}{}^*$ | $S_g{}^*$ | $\alpha_{ent}{}^*$ | $IM_{err}$ |
|---|---|---|---|---|---|
| S | $1.47 \times 10^{-2}$ | 100 | 70 | 0.25 | 0.13 |
| K | 25 | 0.006 | 80 | 0.25 | 0.16 |
| KE | 5250 | 0 | 80 | 0.25 | n/a |

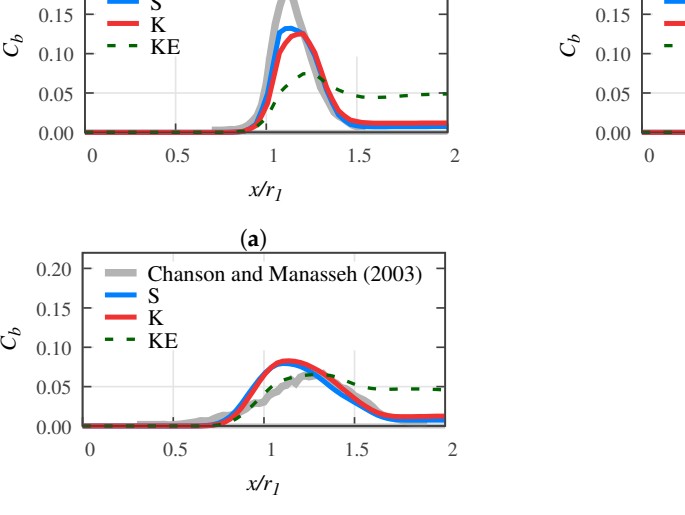

**Figure 5.** Impinging jet—bubble volume fraction prediction for entrainment formulations S, K and KE: (**a**) $z = -0.02$ m. (**b**) $z = -0.03$ m. (**c**) $z = -0.05$ m.

The sensitivity of the SGBM parameters $a_b$, $P_{r0}$, $S_g$, and $\alpha_{ent}$ are analyzed regarding the two entrainment formulations with accurate performance: 'S' and 'K' (see (9) and (10)). Centered on the best quadruplet (Table 2), the $IM_{err}$ error resulting from a variation of each parameter is evaluated and presented in Figure 6.

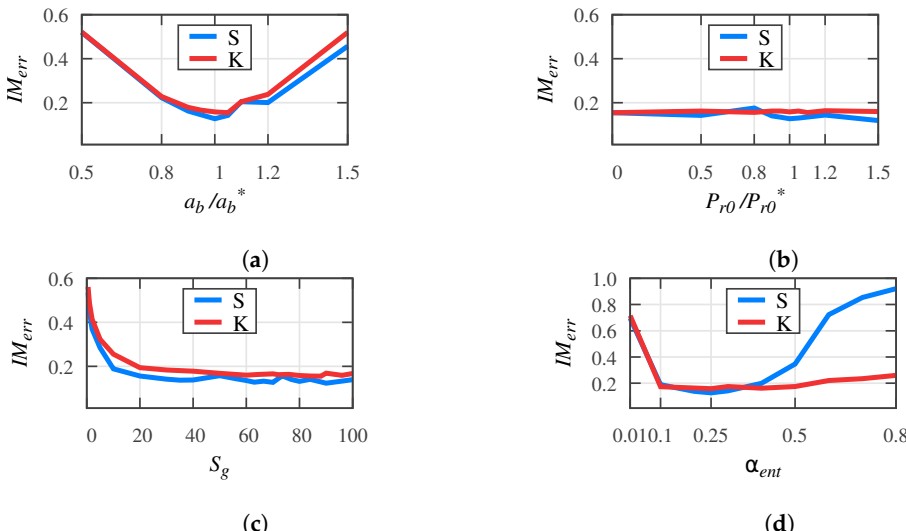

**Figure 6.** Impinging jet—sub-grid model sensitivity to parameters: (**a**) $a_b$. (**b**) $P_{r0}$. (**c**) $S_g$. (**d**) $\alpha_{ent}$.

Both entrainment formulations present similar results. Concerning $a_b$, $IM_{err} < 0.2$ if the variation is inferior to $\pm 20\%$. However, a shift of $\pm 50\%$ conducts to an unsatisfactory $IM_{err} \approx 0.5$.

$P_{r0}$ variations do not change the model accuracy significantly. An apparent improvement is found at the high end of formulation 'S', although the entrainment becomes focused on very few cells, which is non-realistic.

$S_g$ dependence is approximately constant between 20 and 100, with a slight enhancement at the higher values. Nonetheless, for $S_g > 20$, the results are proximate.

Identical and proper $IM_{err}$ is found in both entrainment formulations if $0.1 < \alpha_{ent} < 0.4$. At larger values, the sensitivity to this parameter is extreme for formulation 'S', yet mild for 'K'.

Using the SGBM entrainment formulation 'K' best parameters, a single group bubble population ($r_{b,i} = 0.5$ mm) is preliminarily compared with a ten-group population ($r_{b,i} = \{0.25; 0.34; 0.40; 0.47; 0.55; 0.65; 0.76; 0.89; 1.05; 6.0\}$ mm) that includes all slip velocities of (17). The bubble volume fraction profiles are almost identical. The ten group population shows a slight increase in the maximum value at all profiles and a calculation time 30% larger.

A preliminary test of the bubble population number of groups is conducted in the impinging jet. Figure 7 shows similar behavior for a single group or ten groups with distinct bubble size, though computational time increases by 30%. This result is expected because, in the SGBM, the number of groups and their characteristic radius does not influence the volumetric bubble onset (12). Furthermore, bubble size only controls the bubble slip velocity (16) [34], which is small when compared to the flow mean velocity at the entrainment region vicinity ($w_s \approx V_1/10$). However, once the bubble coalescence and break-up are integrated, the number of groups and bubble size are expected to have the most impact on the transport, break-up and coalescence processes.

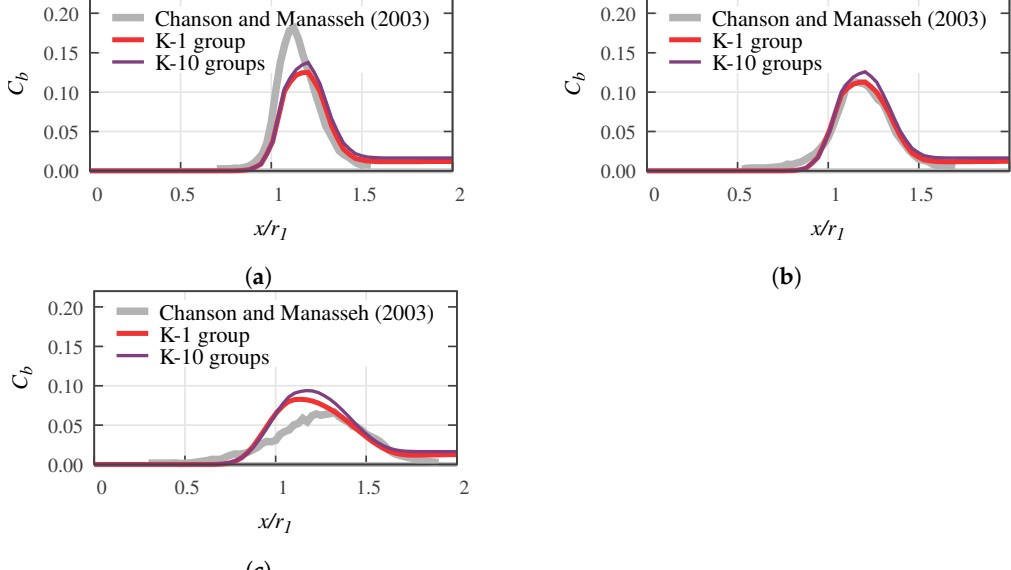

**Figure 7.** Impinging jet—bubble volume fraction prediction for entrainment formulations K with 1 and 10 bubble groups: : (**a**) $z = -0.02$ m. (**b**) $z = -0.03$ m. (**c**) $z = -0.05$ m.

The SGBM volumetric bubble concentration prediction for the impinging jet is very satisfactory. Both entrainment formulations 'K' (10) and 'S' (9) achieve an accuracy of about 15%. The bubble production term threshold ($P_{r0}$) is easily predicted with 'K'; formulation 'S' is less dependent on turbulence yet more sensitive to the entrainment surface location. No acceptable results are found with the formulation 'KE' (11), hence it is not evaluated. Regarding the model parameters, a high dependence is found for the entrainment coefficient $a_b$, whilst the accuracy is equal if $S_g > 20$. Therefore, in the vicinity of the impinging point, bubble advection is more critical than diffusion.

### 3.2. Continuum Aeration in a Spillway Chute

The second case study is continuum aeration in a spillway chute. Free-surface aeration, or self-aeration, is a natural process at an extensive region of the free-surface of high-velocity flows where air bubbles are entrained, and water droplets are formed [45]. Although the physical mechanisms are not entirely understood, the primary cause is the rise of the turbulent kinetic energy generated at the boundaries [14]. Self-aeration occurs when the turbulence at the free-surface is enough to overcome surface tension and bubble buoyancy. The point of inception (PI) is where this phenomenon initiates [46].

Three main aspects of free-surface aeration differ from the local aeration in the impinging jet previously analyzed. First, this process extends over several meters, initiates downstream of the PI and usually continues until the end of the chute. Second, for a fixed spillway discharge, the phenomenon may be considered steady-state spatially and temporally. Third, the energy source for bubble entrainment transport is generated at a boundary. Thus, it may be classified as a boundary layer flow rather than an internal shear flow, such as the impinging jet that has no significant wall effect.

The present section evaluates the combination of the RANS and VOF models with the SGBM to simulate self-aeration and bubble transport at a usual spillway solution: a WES weir [47] followed by a straight chute with a constant slope. Generally, hydraulic structures with such slope and length imply extremely high-velocity flows.

The PI location and the depth-averaged volumetric bubble concentration are evaluated.

### 3.2.1. Numerical Setup

The continuous air entrainment study is focused on the chute flow, specifically the section that includes the PI and the free-surface aeration. Nevertheless, a WES weir is implemented to reproduce the velocity profile at the beginning of the chute (Figure 8).

The chute is 125 m long with a constant slope of 1:1.5 ($\theta \approx 33.7°$), which stays between the typical values for ogee (60°) and side-channel (5–10°) spillways. A sand-grain roughness ($k_s$) of 0.001 m is adopted to simulate concrete roughness. The WES weir design head is 5 m, the upstream platform level is 10 m below the crest, and the resultant linear discharge ($q$) is 24 m$^2$s$^{-1}$. The inlet is 40 m upstream of the crest, and the top boundary is 60 m above the bottom. $l_c$ is the distance to the weir crest along the spillway invert and $l_{PI}$ is the point of inception distance to the weir crest along the spillway invert. $h$ is the water depth and $Y$ is the distance from the bottom measured perpendicular to the spillway invert.

A hexahedral 2D mesh is used due to the hundreds of simulations needed to calibrate the SGBM. The resolution is not uniform. The characteristic edge length ranges from 0.8 m at the top boundary to 0.05 m at the chute. Approximately 30 cells per water depth are found at the point of inception. The mesh contains 199,493 cells. This configuration lacks the typical guidance walls, enhancing the approximation velocities upstream of the weir. Still, no significant effect is reflected in self-aeration.

Regarding boundary conditions, the inlet has a fixed water flowrate; hence, it adjusts the velocity according to the water level on the inside. Additionally, smooth transitions are applied to the inlet velocity profile at the bottom and air–water interface. At the top, a total pressure is combined with a binary velocity condition: zero gradient for outflow and pressure-driven inflow with an absolute angle of −45°. Bottom tangent velocity is null and combined with a turbulence-based roughness wall function. At the end of the chute, inflow is not allowed, and outflow velocity has zero gradient.

Considering the flow is dominated by the boundary layer, the $k$–$\omega$ SST turbulence model is applied due to its better performance near the walls when compared to the $k$–$\varepsilon$ model. The numerical schemes used are: Euler time derivative; limited-linear for the divergence of velocity; linear-upwind for the divergence of $k$ and $\omega$; van Leer for the divergence of $N_b$; linear for the divergence of $\alpha$. The interface compression coefficient ($C_\alpha$) is 0.1. An additional correction is employed to account for mesh non-orthogonality.

Data sampling is performed after a warm-up period where semi-steady flow conditions are reached and approximate time-independence is attained for local fields and global properties, i.e., kinetic energy, turbulent kinetic energy, water level, maximum velocity, and volumetric bubble concentration. The samples are collected in each time-step and time-averaged. Using the RANS model exclusively, the sampling interval is 20 s. For the SGBM calibration, 2.0 s are sampled.

Computing the 2D mesh in an HPC cluster with an Intel Xeon E5-2680 (2.70 GHz) processor using all 16 cores, each run execution time is approximately 1 h per 7 s of simulation time. Applying the SGBM, calculation time increases by less than 5%.

### 3.2.2. Spillway Chute Base Flow

The spillway flow is characterized by practically straight streamlines, though significant contractions are presented immediately upstream of the crest (Figure 8). At the inlet, the crest, and the end of the chute, the depth-averaged velocity is, respectively, 1.6 m s$^{-1}$, 6.5 m s$^{-1}$ and 32.0 m s$^{-1}$ and the depth is 14.80 m, 3.68 m and 0.75 m. At the end of the chute, the flow is still far from the uniform regime.

Air velocity is globally below 0.5 m s$^{-1}$. The water flow drags air at the chute, developing an interface boundary layer with an average thickness of 0.5 m. Upstream of the crest, low water velocity magnitude and gradients inhibit significant turbulence production. Along the chute, bottom friction generates a turbulent boundary layer.

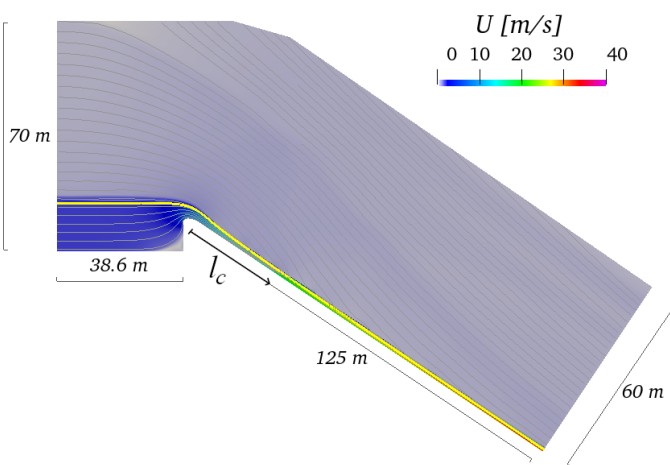

**Figure 8.** Spillway—longitudinal profile: velocity and water surface (yellow line).

The bubble production terms ($P_r$) of formulations 'K' and 'S' are compared. Three profiles perpendicular to the invert – before, near and after the PI (Figure 9)—evidence $\nu_t$ and $k$ similarities. However, $|S|^2$ very large gradients at the air–water interface emphasize $P_r^S$ sensitivity to the entrainment surface location, defined by $\alpha_{ent}$.

After the PI, $P_r^K$ grows approximately linear at the water surface until the end of the chute (Figure 10). This term is exclusively proportional to $k$, which is generated at the bottom, and increases continuously before the uniform regime is attained. Contrarily, $P_r^S$ reaches a maximum after 15 m and decreases downstream. Formulation 'S' behavior at the water surface is considered incoherent with the arise of the bottom turbulent boundary layer. Therefore, it is classified as inadequate and is not applied on the spillway chute case.

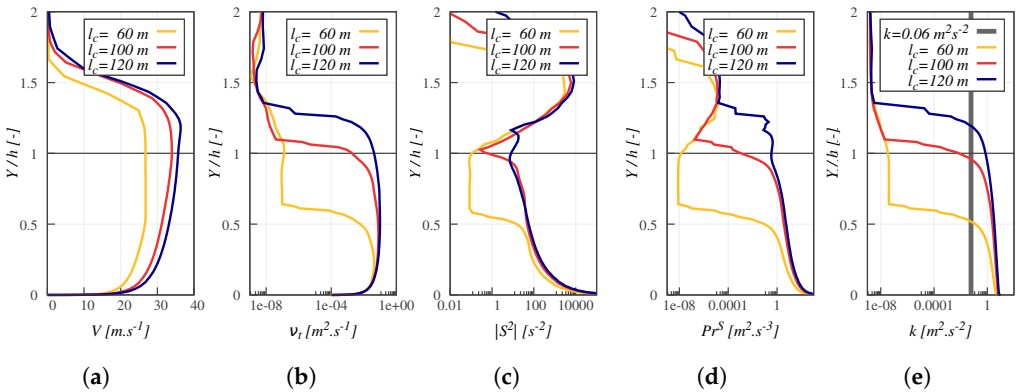

|(a)|(b)|(c)|(d)|(e)|

**Figure 9.** Spillway chute—profiles at $l_c = \{60, 100, 120\}$ m: (**a**) $V$. (**b**) $\nu_t$. (**c**) $|S|^2$. (**d**) $P_r^S$. (**e**) $k$.

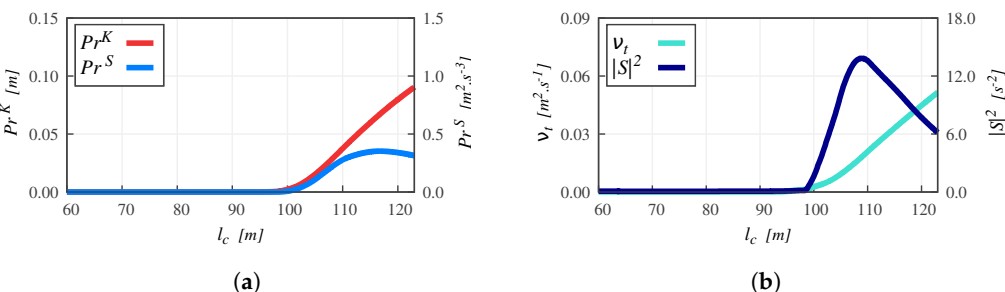

|(a)|(b)|

**Figure 10.** Spillway chute—longitudinal profile at water surface: (**a**) $P_r^K$, $P_r^S$. (**b**) $\nu_t$, $|S|^2$.

### 3.2.3. Sub-Grid Bubble Model Calibration

The SGBM calibration evaluates the model's accuracy and its sensibility to the parameters. It is only performed for the production term formulation 'K', based on $k$ (10), due to the lack of successful results obtained in the previous section.

Considering that the bubble inter-group transfers are neglected to focus on the entrainment process analysis and that the volumetric bubble onset is independent of the population characteristics (8), a single bubble group is modelled. Cain [48] states that the average bubble diameter is larger than 10 mm near the free-surface and decreases towards the bottom to less than 1 mm. Hence, an arbitrary bubble radius ($r_b$) of 10 mm is modelled.

The depth-averaged volumetric bubble concentration (39) along the chute is compared with the analytical solution of Chanson [27], (40). The equation is proposed for a constant slope channel in the gradually varied flow region, with the origin at the PI, as follows:

$$C_{b,mean} = \frac{1}{Y_{90}} \int_{Y=0}^{Y=Y_{90}} C_b \, dY \tag{39}$$

$$\frac{1}{(1-C_{b,e})^2} \ln\left(\frac{1-C_{b,mean}}{C_{b,e}-C_{b,mean}}\right)$$
$$-\frac{1}{(1-C_{b,e})(1-C_{b,mean})} = k' \, s' + K_o \tag{40}$$

where

$$K_o = \frac{1}{1-C_{b,e}}\left(\frac{1}{1-C_{b,e}} \ln\left(\frac{1}{C_{b,e}}\right) - 1\right)$$

$$k' = \frac{w_s \, h_{PI} \, \cos\theta}{q}$$

$$s' = (l_c - l_{PI})/h_{PI}$$

$Y_{90}$ is the characteristic depth where the air concentration is 90%, $h_{PI}$ is the water depth at the point of inception and $C_{b,e}$ is the depth-averaged bubble volumetric concentration in uniform equilibrium flow (41).

$$C_{b,e} = 0.9 \sin\theta \tag{41}$$

The slip velocity is calculated: $w_s(r_b = 10 \text{ mm}) = 0.34 \text{ m s}^{-1}$ (17).

Additionally, after calibration, the simulated PI location is compared against Chanson [49] for smooth chutes (42), expressed as:

$$l_{PI} = 13.6 \, k_s (\sin\theta)^{0.0796} (F_*)^{0.713} \tag{42}$$

$$F_* = q / \left(g \sin\theta \, k_s{}^3\right)^{1/2}$$

The $C_{b,mean}$ and $l_{PI}$ calculation accuracy is evaluated by a mean absolute relative error (43), which, respectively, adopts the '$C_{b,mean}$' and 'PI' superscripts.

$$mean_{err} = \frac{1}{N} \sum_{i=1}^{N} \left|\frac{M-R}{R}\right| \tag{43}$$

where $N$ is the data series number of samples.

The numerically obtained point of inception (PI) is a function of the flow properties, numerical setup and the parameters $P_{r0}$ and $\alpha_{ent}$ that are not calibrated. Chanson [8] suggests that free-aeration triggers if the turbulent velocity ($v'$) exceeds 0.1 to 0.3 m s$^{-1}$ at the surface.

The numerical point of inception location for $v' =$ 0.1, 0.2 and 0.3 m s$^{-1}$ is 103.2, 103.5 and 104.9 m, which corresponds to an error $mean_{err}{}^{PI}$ of 0.8, 1.0 and 2.4%. All the errors are acceptable in practical engineering. The intermediate value is chosen because it corresponds to the best value for the $P_{r0}$ parameter found in the impinging jet for formulation 'K'. The strategy is to reduce the differences in the adopted parameters for different flow types.

Hence, in the spillway chute, the adopted criteria to locate the inception point (PI) are $v'(l_c = l_{PI}) = 0.2$ m s$^{-1}$, which correspond to $k(l_c = l_{PI}) = 0.06$ m$^2$s$^{-2}$ and $P_{r0}{}^K(l_c = l_{PI}) = 0.006$ m. The PI location is 103.5 m, matching Equation (42)'s prediction of 102.4 m accurately, with an error $mean_{err}{}^{PI}$ of 1.0%. At the PI, the water depth ($h_{PI}$) is 0.83 m , the average velocity is 28.9 m s$^{-1}$ and the maximum velocity is 34.4 m s$^{-1}$.

Nevertheless, the VOF interface thickness affects the PI location. Therefore, the $C_\alpha$ parameter of the compression term (6) may play an important role.

In the spillway chute case, the following four parameters are kept constant: $P_{r0}{}^K(v' = 0.2$ m s$^{-1}) = 0.006$ m (8); $\alpha_{ent} = 0.5$; $C_{b,max} = 1$; $\alpha_{det} = 0.0001$.

The calibration process focused on the remaining and most relevant parameters:

- $a_b$—bubble onset coefficient (8);
- $S_g$—Schmidt number for gases in water (18).

Five hundred combinations of $a_b$ and $S_g$ are tested, and the resultant $mean_{err}{}^{C_{b,mean}}$ is used to find the optimal parameter combination ('*' index), presented in Table 3. The SGBM calibration process is similar to the one adopted for the impinging jet, described in Section 3.1.3. The main difference is that only two parameters are calibrated; therefore, step 1 is not performed. Step 2 and 3 implied, respectively, two hundred and thirty and three hundred quadruplet tests.

The best performance shows a good agreement with Chanson [27], presenting a $mean_{err}{}^{C_{b,mean}}$ error of 8.5%, which is completely appropriate in the analysis of hydraulic structures. Therefore, the entrainment formulation 'K' proves its ability to model self-aeration in spillway chutes. The bubble volumetric concentration at a chute stretch containing the point of inception and downstream is shown in Figure 11. The bubble mean volumetric concentration downstream of the PI is plotted against the analytical solution of Chanson [27] (40) in Figure 12. The two curve discontinuities are due to the discrete entrainment surface concept (see Appendix B). In other words, entrainment occurs at a single layer of cells that follows the water surface. However, the water surface slope is not entirely aligned with the mesh cell faces, resulting in occasional steps in this entrainment layer.

**Table 3.** Spillway—calibration of sub-grid model: best parameters. Entrainment formulation 'K'.

| $a_b{}^*$ | $P_{r0}{}^*$ | $S_g{}^*$ | $\alpha_{ent}{}^*$ | $mean_{err}{}^{C_{b,mean}}$ |
|---|---|---|---|---|
| 700 | 0.006 | 1.5 | 0.5 | 0.085 |

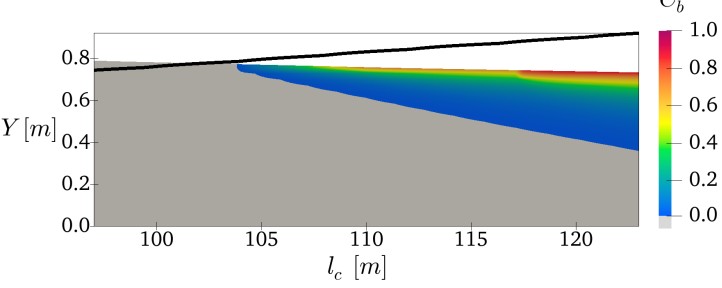

**Figure 11.** Spillway chute—bubble volumetric concentration. Black line is $k = 0.06$ m$^2$s$^{-2}$.

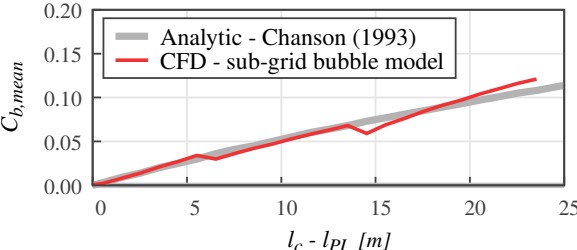

**Figure 12.** Spillway chute—bubble mean volumetric concentration against Chanson [27] analytical solution.

Finally, the sub-grid bubble model sensibility to the parameters $a_b$ and $S_g$ (shown in Figure 13) is analyzed regarding $mean_{err}{}^{C_{b,mean}}$ (40).

Concerning $a_b$, $mean_{err}{}^{C_{b,mean}} < 0.2$ in the range $[0.65\ a_b{}^*, 1.5\ a_b{}^*]$. Furthermore, a positive shift does not increase the error significantly. $S_g$ dependence is considerable. A satisfactory $mean_{err}{}^{C_{b,mean}} < 0.2$ is found between 1.1 and 2. Unlike the impinging jet, in the spillway, the SGBM sensitivity is higher to $S_g$ than $a_b$. Hence, suggesting that the phenomenon is dominated by bubble diffusion and that the entrainment layer cells are at the maximum bubble fraction capacity.

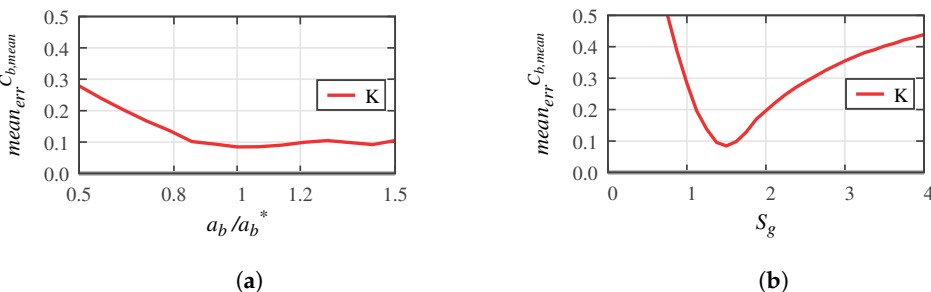

**Figure 13.** Impinging jet—sub-grid model sensitivity to parameters: (**a**) $a_b$. (**b**) $S_g$.

## 4. Discussion

### 4.1. RANS Model Constraints

The sub-grid bubble model SGBM performance is first and foremost conditioned by the RANS model capacity to replicate the flow.

Despite the spillway chute being correctly simulated, difficulties arise in the reproduction of the submerged shear layer and momentum diffusion in the impinging jet. When applying the VOF method, the mesh should be fine enough to reproduce the individual bubbles [42], which is unfeasible for the vast majority of hydraulic structures due to their dimensions.

Furthermore, in the VOF method, the air–water interface is not entirely defined. Instead, it is a region with a thickness that heavily depends on the mesh resolution and the numerical settings. Therefore, some uncertainty rises on the water surface exact position, which significantly impacts the calculation of the air entrainment.

Turbulence modelling is also critical to the entrainment process, especially at the interface and submerged shear layer. Different turbulence model and numerical setup may alter the outcome significantly. However, validation is problematic because laboratory or prototype data are practically inexistent [3].

### 4.2. Air Entrainment Formulations Performance

The volumetric bubble concentration prediction of the entrainment formulation 'K' (10) [17], based on the turbulent kinetic energy ($k$), is very good for local and continuum aeration. Moreover, the threshold to initiate the bubble onset ($P_{r0}$) is easily predicted, related to $v'$ or $k$. Along with other authors [40,45,50,51], Ma et al. [17] consider that, for lower jet velocities (inferior to 4 to 6 ms$^{-1}$), the leading cause of entrainment is jet

roughness. The latest author also assumes that the jet roughness is proportional to $k$. In spillway chute self-aeration, a similar hypothesis is adopted: the rate of air entrainment is proportional to the size of surface perturbations caused by the rise of the turbulent kinetic energy ($k$) from the bottom.

The entrainment production term formulation 'S' (9) [11], based on $\nu_t$ and $|S|^2$, is effective in shear flows and less dependent on turbulence than formulation 'K'. However, it does not comply with self-aeration. This term is related to $k$ production (21). Therefore, it may be useful if the inflow has minor surface perturbations and large velocity gradients at the impinging point (e.g., plunging waves). Yet, this formulation fails if the inflow turbulent kinetic energy ($k$) is dominant. Furthermore, the threshold ($P_{r0}$) foresee is intricate.

No acceptable results are found with the entrainment formulation 'KE' (11), based on $k$ and $\varepsilon$ [28], applied to the impinging jet. Hence, it is not evaluated in the spillway. Whatever parameter combination set, aeration is not confined to jet vicinity. This approach applies the theoretical concept of turbulent length scales to estimate the surface disturbances as a function of the turbulent eddies. A hypothesis is that this formulation does not comprise the vast range of time and length scales involved. However, the conducted study is insufficient to conclude its efficiency. Moreover, this approach is more sensitive to turbulence modelling because it depends on $k$ and $\varepsilon$. The mathematical model equations and the numerical implementation are also essential. The adopted modelling framework may not be adequate for this approach. Further investigation is recommended, considering Lopes et al. [25] achieved good results in a stepped spillway.

### 4.3. Sensibility to Parameters

The sub-grid bubble model calibration classifies $a_b$ and $S_g$ as the most relevant parameters, followed by $P_{r0}$ and $\alpha_{ent}$.

The optimal bubble onset coefficient ($a_b$) for the spillway is approximately thirty times larger than for the impinging jet, respectively, 700 and 25. Similarly, in their numerical study, Ma et al. [17] found an $a_b$ for a hydraulic jump six times larger than for the impinging jet. Further investigation is needed to scrutinize this behavior.

The Schmidt number for gases in water ($S_g$) typically ranges from 0.4 to 0.9 in hydraulic structures [52,53]. However, in the impinging jet, good results are found if $20 < S_g < 100$. In a similar impinging jet setup, Shi et al. [54] estimated $S_g$ between 30 and 40 at $z = -0.14$ m in the turbulent shear layer vicinity. The author suggests that inertial forces, enhanced by gravity, dominate the high-velocity region, which can justify such a high value. Gualtieri et al. [53] pointed out that $S_g$ may not be constant in the whole domain of non-isotropic flows, which is the case of the impinging jet. Furthermore, in this region, a hypothetical over-estimation of the turbulent viscosity by the $k$–$\varepsilon$ turbulence model can force the $S_g$ to increase to compensate for it (see Equation (18)). Detailed turbulence data are needed to address this issue properly. In self-aeration, Chanson [45] indicates that $S_g$ increases with larger Reynolds numbers, and depends on the mean air concentration. In the spillway chute, good results are found if $1.1 < S_g < 2$, which is within the literature values presented by Chanson [49].

The threshold of the bubble onset production term ($P_{r0}$) is essential to locate the entrainment, especially in self-aeration. Chanson [8] turbulent velocity criteria ($0.1 < v' < 0.3$ m s$^{-1}$) proved to be effective.

The entrainment surface layer location is defined by $\alpha_{ent}$, which refers to a water volume fraction value. In the spillway chute, the standard VOF interface value of 0.5 delivers accurate results. However, the impinging jet is more sensitive. Good results are found if $0.1 < \alpha_{ent} < 0.4$. Lopes et al. [19] suggest $\alpha_{ent} = 0.3$. Additionally, this author warns that $\alpha_{ent}$ may significantly affect the bubble onset location and intensity, although the entrainment surface shape does not change considerably. These facts are confirmed. Therefore, $\alpha_{ent}$ is entirely dependent on the numerical setup. Special attention must be given when there is a rupture of the water surface, e.g., plunging flows.

In local aeration, a higher dependence is found for $a_b$ than for $S_g$. Hence, in the vicinity of the impinging jet, bubble advection is predominant over diffusion. Oppositely, continuum aeration shows a diminished sensitivity to $a_b$ yet very high to $S_g$. Therefore, diffusion prevails upon entrainment quantity. The physical processes confirm these conclusions.

*4.4. Overall Appreciation*

Incorporating the sub-grid bubble model (SGBM) provides a breakthrough improvement over the RANS model results. It allows prediction of previously non-detected aeration with good accuracy, exempting the tremendous computational resources necessary to represent an individual bubble.

Additionally, from the three entrainment formulations tested in the present mathematical model, the entrainment formulation depending exclusively on the turbulent kinetic energy, based on the approach of Ma et al. [17], is the only one found to be appropriate for local and continuum aeration. The range of flow applications of the presented framework is extended to local aeration due to an impinging jet and self-aeration in chutes, which adds to the breaking waves studied by Shi et al. [11].

Moreover, the unprecedented sensitivity analysis of the SGBM parameters exposes the reliability of this framework to two different flows. Calibration is straightforward in self-aeration but more difficult for local aeration. The discrepancy is primarily due to the RANS model ability to reproduce the water flow, which is not as good in the impinging jet as in the spillway chute.

The air entrainment process is very complex in both cases, involving an extended range of flow structures and bubble sizes. The presented framework cannot replicate these phenomena because it requires an extremely high mesh resolution and more advanced Navier–Stokes equation solving procedures than RANS. In the spillway chute, the exact shape of the water surface instabilities is not reproduced, nor the foam structures or air–water projections present in highly-aerated flows, which constitute a complex multiphase region [45]. In the impinging jet, for slightly higher velocities than considered in the present study ($V_1 > 4$ to $12$ ms$^{-1}$), the dominant air entrainment mechanism is the formation of an elongated air cavity and its subsequent breakup [51], which are not reproducible due to the time and length scale involved.

Another particular aspect is turbulence isotropy. Despite being common to use RANS with two-equation turbulence models (e.g., $k$–$\varepsilon$, $k$–$\omega$ SST), both analyzed flows are considered anisotropic. At the chute water surface, the predominant vortexes axes are aligned with the transversal direction, and the entrainment is mainly due to the velocity fluctuations perpendicular to the surface [14]. On the other hand, in the impinging jet helicoidal trajectories of small bubbles are observed around the jet centerline [39,40], and the transport of bubbles away from the shear zone by large vortices with main axis perpendicular to the jet direction [45,51]. Especially in the jet, a turbulence model that complies with high degrees of anisotropy, such as the Reynolds stress equation turbulence model(RSM), could improve the bubble dispersion simulation [55].

The presented coupling of the RANS and VOF models with a SGBM implies three limitations to be considered when selecting an application. First, bulking is not modelled, which is particularly relevant in heavy aerated flows. Second, the volumetric bubble concentration must stay below 20%, although a few cells can exceed this value without compromising the solution. Third, air may entrain from concurrent sources (air pockets from the RANS model and bubble from the SGBM), overdoing the flow buoyancy. In addition, the break-up of large air pockets into bubbles is not possible. Thus, the modelling of strong impinging flows such as intense wave-breaking is excluded.

## 5. Conclusions

A sub-grid bubble model is coupled with a RANS model, which includes the VOF method, to predict the entrainment of an air bubble population in two different hydraulic structures flows: an impinging jet and along a spillway chute. Previously non-detected

entrained air is now simulated with good accuracy, and the additional computation cost is marginal. Therefore, the framework, based on Shi et al. [11] approach is considered valuable and efficient for simulating local and continuum aeration, matching engineering standards.

Three distinct air entrainment formulations are evaluated, yet only the one depending exclusively on the turbulent kinetic energy [17] proved to apply to different types of flow. The framework reliability is exposed by an unprecedented sensitivity analysis of four parameters defined by the user. The sensitivity to the bubble onset threshold ($P_{r0}$) and the entrainment surface water volume fraction ($\alpha_{ent}$) is considered low, and both are easily predicted. The bubble onset coefficient ($a_b$) must be calibrated according to the flow type. The Schmidt number for gases in water ($S_g$), which is fundamental for bubble diffusion, may be difficult to foresee in the impinging jet shear layer.

The simulated impinging jet volumetric bubble concentration matches the laboratory data profiles of Chanson and Manasseh [15] with a combined relative error of 16%. The continuum entrainment at the spillway chute achieves a relative error of 9% against the prototype-based analytical solution of Chanson [27] for the depth-averaged volumetric bubble concentration. Moreover, the inception point corresponds to Chanson [49] expression for smooth chutes with a relative error of 1%.

Local entrainment is restrained by the RANS model ability to reproduce impinging water flows. Furthermore, it is more susceptible to the sub-grid bubble model parameters. Continuum aeration prediction is more reliable.

The application of the presented framework is limited to flows where the volumetric bubble concentration does not exceed 20%, and without significant bulking or the break-up of large air pockets into bubbles. Moreover, turbulence modelling is critical to the bubble onset calculation.

Further development and evaluation in different flows are crucial to validate this tool for hydraulic structures engineering purposes.

**Author Contributions:** Conceptualization, L.S.M. and J.L.L.; methodology, L.S.M., J.L.L. and M.T.V.; software, L.S.M.; validation, L.S.M., J.L.L. and M.T.V.; formal analysis, L.S.M.; investigation, L.S.M.; resources, L.S.M., J.L.L. and M.T.V.; data curation, L.S.M.; writing—original draft preparation, L.S.M.; writing—review and editing, L.S.M., J.L.L. and M.T.V.; visualization, L.S.M.; supervision, J.L.L. and M.T.V.; project administration, J.L.L.; funding acquisition, L.S.M. and M.T.V. All authors have read and agreed to the published version of the manuscript.

**Funding:** The authors acknowledge the following institutions for their funding and support: Fundação para a Ciência e a Tecnologia (FCT), Portugal—first author PhD Grant SFRh/BD/99815/2014; Laboratório Nacional de Engenharia Civil (LNEC), Portugal—first author PhD Grant co-funding. This work is partially funded under the 3rd Multi-Thematic Joint ERANet-LAC Call 2017/2018 (grant ERANet17/ERY-0222).

**Institutional Review Board Statement:** Not applicable.

**Informed Consent Statement:** Not applicable.

**Data Availability Statement:** Not applicable.

**Acknowledgments:** The authors acknowledge the following institutions for their support: Laboratório Nacional de Engenharia Civil (LNEC), Portugal—first author PhD Grant host; Instituto de Hidráulica Ambiental (IH Cantabria), Spain—first author host; Infraestrutura Nacional de Computação Distríbuida (INCD) funded by FCT and FEDER (European Regional Development Fund) under the project 01/SAICT/2016 nº 022153, Portugal—computational resources.

**Conflicts of Interest:** The authors declare no conflict of interest.

**Abbreviations**

The following abbreviations are used in this manuscript:

| | |
|---|---|
| CFD | Computational Fluid Dynamics |
| DNS | Direct Numerical Simulations |
| DES | Detached Eddy Simulation |
| ES | Entrainment surface |
| HPC | High-performance computing |
| LS | Level-Set method |
| PI | Point of inception |
| RANS | Reynolds-average Navier–Stokes equations |
| SGBM | Sub-grid bubble model |
| SST | Shear Stress Transport |
| TL | Top layer |
| USBR | United States Bureau of Reclamation |
| VOF | Volume-of-fluid method |
| WES | Waterways Experiment Station, United States Army Corps of Engineers |

**Appendix A. Numerical Implementation**

The sub-grid bubble model (SGBM) is encompassed in an OpenFOAM® library that compiles separately. The base solver (*interFoam*) code had minor changes: a bubble buoyancy term is added to the RANS momentum Equation (2), and two calls initiate and re-calculate the SGBM. All data and calculations are comprised in a 'bubble model' class object created according to the parameters and settings of an input file.

At each time-step, a set of operations is performed sequentially:

(i) identify the entrainment surface cells (A4);
(ii) inter-group transfers;
(iii) bubble entrainment (14);
(iv) bubble transport (15);
(v) turbulence;
(vi) momentum Equation (2).

**Appendix B. Air-Bubble Entrainment Surface Detection**

A new method locates bubble entrainment at a layer of cells defined by a VOF fraction value ($\alpha_{ent}$). Relying only on fields operations instead of geometric calculations, it proves to be effective and very fast. Bubble entrainment occurs at a single layer of cells—named top layer (TL)—just below an iso-surface (ES) defined by a specified VOF fraction value ($\alpha_{ent}$). The following steps determine the entrainment layer:

1. create a $\alpha_{ent}$ comparative binary function (A1) for all domains.

$$\psi^{cell} = \begin{cases} 0, & \alpha < \alpha_{ent} \text{ (above ES)} \\ 1, & \alpha \geq \alpha_{ent} \text{ (below ES)} \end{cases} \tag{A1}$$

2. Linear interpolate $\psi^{cell}$ to the faces ($\psi^{face}$) and calculate the face area-weighted average for each cell (A2). $A_f$ is the cell face area. A second binary function arises (A3) that selects the top layer (TL) of cells below ES.

$$\psi_{\bar{f}}^{cell} = \frac{\sum_f^{faces} \psi^{face}{}_f A_f}{\sum_f^{faces} A_f} \tag{A2}$$

$$\psi_{TL}^{cell} = \begin{cases} 0, & \psi_{\bar{f}}^{cell} \geq 1 \text{ (below TL)} \\ 1, & \psi_{\bar{f}}^{cell} < 1 \text{ (TL and above)} \end{cases} \tag{A3}$$

3.  Finally, a domain mask (A4) is found, which identifies the single layer of cells where bubble entrainment takes place.

$$mask_E^{cell} = \psi^{cell} \; \psi_{TL}^{cell} = \begin{cases} 0 & \text{(not entrainment layer cells)} \\ 1 & \text{(entrainment layer cells)} \end{cases} \tag{A4}$$

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
