# Peer review of "Is the Volume-of-Fluid Method Coupled with a Sub-Grid Bubble Equation Efficient for Simulating Local and Continuum Aeration?"

_water, doi:10.3390/w13111535_

Round 1

Reviewer 1 Report

This is a very ambitious manuscript, as is the research that it represents.  Overall, I believe that it should be published but there are some issues that should be addressed.  First of all, the authors should get someone who is an English speaker to edit the manuscript for grammar.  There are many places with writing not in complete sentences, apparent incorrect choices of words (where words similar to the one selected was apparently intended) and other obvious errors.

With the ambitious objectives of the study, It is perhaps not surprising that the work is somewhat incomplete still and I am not so sure that all conclusions are justified.  However, there is some recognition of the limitations in the manuscript and I do not intend to make this a harsh criticism.  There are so many choices (calibrations) that go into the modeling and little available data that is sufficiently detailed against which to assess the results one should not be overconfident that the "correct" approaches have been identified.  Just as one instance, it is somewhat difficult to understand why the turbulent Schmidt number should vary so greatly between the two flow cases considered.    I also wonder why for the simulations for the impinging jet, an intermediate discretization, W32, appears to be the best choice.  That is not typically the type of result that should be expected and suggests to me that there may be another numerical issue involved.  I also understand that the choice of a single bubble size with no interactions makes the numerical application easier but this approach may leave out sufficient physics so that the particular calibration approaches may not be adequate with a more complete model formulation.  Overall, just be a little more direct about the limitations of various modeling approaches and assumptions. For example, in the chute flow simulation, the point of bubble inception is apparently predicted to a very high accuracy of a little over one percent.  However, the definition used in the modeling uses a criteria that was suggested by others to vary by as much as a factor of three and the choice in the modeling was apparently to simply select an arbitrary vale within this range.  The factor of three variation in the turbulent velocity perhaps means that using that as a criterion may be suspect.

A couple of minor editorial comments.  Use of u' for turbulent velocity seems nonstandard to me.  line 781, there is a reference with a question mark.  Usse of alpha although with different subscripts for two completely different quantities is somewhat confusing

Reviewer 2 Report

This is an interesting article presenting new methods for incorporating more detail into the modeling of aeration processes, while not incurring extraordinary computational costs that would occur with grids fine enough to resolve individual bubbles.  My comments are primarily provided in the marked PDF.  There are a number of terms that apparently have not been translated well and need review or adjustment.  They make the meaning of some passages difficult to determine.

Reviewer 3 Report

The study concerns implementation of the OpenFOAM toolbox. The Reynolds-averaged Navier–Stokes equations and the volume-of-fluid method were coupled with a sub-grid bubble population equation, comprising entrainment and transport. The state of the art should clearly show the knowledge gaps identified. The authors do not explain well, where is the novelty of the distinguished method. Please reason both the novelty and the relevance of your paper goals. The original developments have to be properly described and reasoned. All borrowed elements and parts have to be properly summarised and references to the original works provided.
